# Synergistic Activation of Toll-Like and NOD Receptors by Complementary Antigens as Facilitators of Autoimmune Disease: Review, Model and Novel Predictions

**DOI:** 10.3390/ijms21134645

**Published:** 2020-06-30

**Authors:** Robert Root-Bernstein

**Affiliations:** Department of Physiology, Michigan State University, East Lansing, MI 48824, USA; rootbern@msu.edu

**Keywords:** Toll-like receptors, nucleotide-binding oligomerization domain-containing proteins, TLR, NOD, synergy, molecular mimicry, anti-idiotype, bystander activation, antigenic complementarity, complementary antigens, theory, autoimmunity, innate immunity

## Abstract

Persistent activation of toll-like receptors (TLR) and nucleotide-binding oligomerization domain-containing proteins (NOD) in the innate immune system is one necessary driver of autoimmune disease (AD), but its mechanism remains obscure. This study compares and contrasts TLR and NOD activation profiles for four AD (autoimmune myocarditis, myasthenia gravis, multiple sclerosis and rheumatoid arthritis) and their animal models. The failure of current AD theories to explain the disparate TLR/NOD profiles in AD is reviewed and a novel model is presented that explains innate immune support of persistent chronic inflammation in terms of unique combinations of complementary AD-specific antigens stimulating synergistic TLRs and/or NODs. The potential explanatory power of the model is explored through testable, novel predictions concerning TLR- and NOD-related AD animal models and therapies.

## 1. Introduction

Most research concerning the initiation of autoimmune diseases focuses on the role of the adaptive immune system in carrying out a specific attack upon some host autoantigen. Similarly, since the immune system is assumed to be programmed to avoid attacking host antigens, most theories of autoimmune disease (AD) assume that defects in adaptive immunity are responsible for the loss of tolerance that precedes autoimmune diseases. Increasingly, however, it has become apparent that the innate immune system must first be activated in order to stimulate and sustain the adaptive autoimmune response, shifting some of the question of what goes awry during the induction of AD to the innate system. Thus, many investigators have been searching for innate “defects…[and] abnormalities that have been related to inflammatory disorders” [1].

I have been working for some time on a theory of autoimmunity that turns the problem on its head, starting with the assumption that there is no defect in tolerance nor defect in the functioning of the immune system (adaptive or innate) during the induction of AD. My theory assumes instead that the immune system functions in autoimmunity just as it was designed to function. AD results not from a defect or mistake, but from an unusual concurrence of antigenic stimuli that synergistically activate both the innate and adaptive immune systems to produce the unusual phenomena that we observe in AD. The incidence of an AD should therefore be a function of the probability that a genetically susceptible individual will encounter such a synergistic set of antigenic stimuli. An adequate theory of AD should be able to calculate those odds.

This study reviews one aspect of that unusual activation—toll-like receptor (TLR) and nucleotide-binding oligomerization domain-containing proteins (NOD) activation and signaling—and provides a model to explain its function in initiating autoimmune disease. This model is applied specifically to four AD and their animal models: autoimmune myocarditis, myasthenia gravis, multiple sclerosis and rheumatoid arthritis. The implications for the model are then extrapolated to several other AD and a series of experimentally testable predictions are then made that can differentiate the proposed model from other theories of how innate immunity supports AD pathogenesis.

## 2. Toll-Like Receptor Activation in Autoimmune Disease

### 2.1. Normal Innate Immune Function

The innate system is basically comprised of several layers [1]. The first consists of physical and chemical barriers to infection such as the epidermis, mucus, saliva, tears and the broadly acting antimicrobial proteins such as mucins, lysozyme and lactoferrin that these contain. The second consists of cellular components such as phagocytes (macrophages, neutrophils), dendritic cells and mast cells that nonspecifically sample possible antigens and respond with the release of cytokines. The cytokines can be considered the third layer of innate immunity, acting as messengers between innate and adaptive cells and regulating their activation or deactivation. Finally, there is the complement system, which mediates the biochemical destruction of antigenic material by appropriately activated immune system cells. In general, the cellular components of the innate immune system are considered to be the most important mediators of autoimmune processes because these control the cytokine production that stimulates the chronic inflammatory environment driving AD pathogenesis.

Cellular recognition of potential antigens is carried out by pattern recognition receptor (PRR)-bearing cells that recognize pathogen-associated molecular patterns (PAMP) on microbes. PRR express TLR and NOD as well as NOD-like receptors (NLR) that each recognize a particular class of microbial PAMP. Binding activates production of various inflammatory cytokines by means of either the TRIF (and/or TIRAP) or MyD88 pathways [2]. TRIF stands for “Toll/IL-1 receptor–domain-containing adaptor protein inducing INF-α activators” and, as its name says, the TRIF pathway results in the production of interferons (INF). TIRAP is the Toll/IL-1 receptor–domain-containing adaptor protein and activates a pathway releasing interleukins. MyD88 is the myeloid differentiation primary response protein 88, which activates the release of NF-κB (nuclear factor kappa-light-chain-enhancer of activated B cells). TLRs 3, 4 and 6 activate TRIF and therefore cellular (Th1) immunity; human TLRs 1, 2, 4, 5, 7, 8, 9 and 10 activate MyD88 and therefore antibody-mediated (Th2) immunity [3,4]. NOD1 and NOD2 activation (though not all NLR) also leads to MyD88 activation and production of NF-κB [5,6].

The role of TLR and NOD1 and NOD2 in supporting AD pathogenesis is particularly well-established and therefore a brief introduction to TLR and NOD1 and NOD2 specificities is in order. Essentially, different TLR and NOD have evolved to recognize specific classes of compounds produced exclusively by microbes and parasites. Presence of such compounds indicates the need to stimulate an immune response. Because little work has been done on NLR more generally with regard to AD, I focus mainly on the roles of NOD1 and NOD2 here [7] and do not refer further to NLR more generally. NOD1 and NOD2 are two members of a larger group of NLR that help the innate immune system recognize and respond to conserved motifs in bacterial peptidoglycans (PGN). NOD1 recognizes γ-D-glu-meso-diaminopimelic acid (iE-DAP) dipeptide-containing antigens that are found in PGN of all Gram-negative and certain Gram-positive bacteria. NOD2 recognizes PGN containing *N*-acetyl-muramyl-D-alanyl-isoglutamine (MDP)-containing peptides derived from bacterial cell walls of both Gram-positive and Gram-negative bacteria, and it also single-stranded RNAs from viruses [6].

TLR have a broader range of specificities that extend to pathogens other than bacteria. TLR1 recognizes triacyl lipopeptides; TLR2, diacyl lipopeptides and glycolipids; TLR3, double-stranded RNA and poly-I:C, an artificial double-stranded RNA-like polymer used as an adjuvant; TLR4 recognizes lipopolysaccharides (LPS), heat shock proteins, fibrinogen and related proteins; TLR5, flagellins; TLR6, diacyl lipoproteins; TLRs 7 and 8, single-stranded RNA; TLR9, viral and bacterial (CpG) DNA, which differs from eukaryotic DNA in having unmodified bases singularly characteristic of microbes; and TLR10, retroviral proteins. Microbes often express antigens that activate multiple TLR, but any well-defined Ag is likely to activate only one TLR in the innate immune system and therefore will induce only one type of cytokine response. TLRs 3, 4 and 6 can activate the TRIF pathway resulting in interleukin production and a Th1 (or cellular) immune response. The other TLR along with TLR4 activate the MyD88 pathway that results in production of inflammatory cytokines and production of a Th2 (antibody) response. TLR can also be grouped by whether they exist on the cell membrane (TLRs 1, 2, 4, 5, 6) or within cells on endosomal membranes (TLRs 3, 7, 8, 9 and 10) (Figure 1). Notably, the former grouping recognize mainly bacterial antigens; the latter group, mainly viral ones. For the latter group to be active, the antigen must first be processed by phagocytosis before presentation to the TLR.

TLR and NOD activation are particularly relevant to understanding AD pathogenesis because most investigators currently believe that AD are triggered by microbial antigens, particularly viruses and bacteria. Depending on which types of microbes trigger the AD, a different set of TLR will be activated. One outstanding question is whether there is a single set of TLR that must be activated to induce all AD or whether different AD involve activation by different sets of TLR. Clinical and experimental studies suggest the latter as will be demonstrated in detail below and summarized in Table 1. Another outstanding question is how TLR activation is maintained throughout the course of AD, which are almost always chronic in nature since there is no evidence in any AD of ongoing viral or bacterial infection. Indeed, the definition of an AD is that it be caused by the immune system itself rather than by active infection. Hence, once the triggering infections are gone, what maintains ongoing innate activation? Evidence again suggests that this continued activation is maintained by host antigens, though the precise nature of these antigens and their ability to activate TLR are not clear [3,8,9,10,11]. The relationship between the activating triggers of AD and the host antigens that continue to drive innate activation throughout the course of the AD is one of the key issues to be addressed below.

### 2.2. Evidence for an Essential Role of Innate Immunity in Autoimmune Disease

All ADs are characterized by ongoing innate activation and chronic inflammation from the appearance of symptoms throughout the disease process [3,8,9,10,11]. The mechanisms by which innate immunity support AD development are, however, far from clear [12,13,14]. Innate activation invariably precedes adaptive immunity in animal models of AD and involves antigen presenting cells such as dendritic cells, macrophages—and in the central nervous system—microglial cells that often attract natural killer (NK) cells. Dendritic cell activation results in type 1 interferon (IFN) stimulation and IFN regulatory factor (IRF) dysregulation, which seems to characterize most AD [15,16]. All of these cell types express various combinations of TLR and NOD (or NLR) that are upregulated as disease is initiated [13,14]. However, the sets of TLR and/or NOD that are activated vary, as will be explored more fully below, from one AD to another in both animal models and human patients suggesting that there is no single innate pathway that is either necessary or sufficient to induce an AD. In some cases, a Th1 response is predominant while in others a Th2 response dominates. In allergic, gastrointestinal and central nervous system AD, mast cells and innate lymphoid cells have also been implicated in disease initiation and progression, controlling the adaptive response through their release of interleukins and tumor necrosis factors [17,18]. In addition, excessive and uncontrolled complement activation is also a universal feature of AD, mediating the actual cellular destruction that accompanies disease [13,19]. Increases in circulating immune complexes (CIC) correlate with increases in complement activation suggesting that antibodies may function to drive ongoing innate immune responses in AD [20,21]. CIC-driven complement activation and antibody binding to TLR and NOD are mechanisms to be discussed more fully below.

Given that there are over one hundred known or suspected AD at present, it is clearly impossible to explore the roles of TLR and NOD in all of these here. For the sake of manageability, this review will therefore limit itself to discussing TLR and NOD involvement in four quite different sets of AD and their respective animal models: autoimmune myocarditis, which attacks heart muscle proteins; multiple sclerosis, which destroys the myelin sheath of nerves; myasthenia gravis, which targets the acetylcholine receptor; and rheumatoid arthritis, which destroys the cartilage in joints. Each disease has well-defined patterns of TLR-NOD activation in the human and animal models and can be induced in animals by well-defined antigens for which the innate activation profiles have been established. 

Human autoimmune myocarditis is characterized by activation and overexpression of TLRs 2, 4, 7 and 8 [22,23] (Table 1). The human disease is modeled by experimental autoimmune myocarditis (EAM), which can be induced in at least three ways. One is to inoculate mice with apoptotic cardiomyocytes along with LPS [24]. Another model involves infecting mice with coxsackie virus B3 (CVB3) cultured in cardiac tissue (and therefore containing material from apoptotic cardiomyocytes including cardiac myosin). The third model involves inoculating animals with either cardiac myosin or its very close bacterially derived molecular mimic, the M protein from group A *Streptococci* (GAS), in complete Freund’s adjuvant (CFA: a mixture of killed *M. tuberculosis* in mineral oil). In all three forms of EAM, TLR2 and TLR4 are activated [25,26,27]. TLR7 activity is also essential to myosin-CFA induced EAM, but not TLR3 or TLR9 [28]. In sum, EAM, like its human counterpart, is characterized by activation of TLR2, TLR4 and TLR7.

As just noted, most animal models of AD require a combination of more than one agent. In describing the induction of animal models of AD, it is usual to identify one of the agents as the ”antigen” and the other as the “adjuvant”. The antigen is usually an agent that in some way mimics a host molecule that is the target of the AD. This may be, as in EAM, cardiac myosin from a different species of animal (a so-called “autologous antigen”) or it may be a molecular mimic of a host molecule, such as the GAS M protein that is very similar to cardiac myosin. The antigen is the molecule that targets the autoimmune process to a specific host tissue. The second agent—or adjuvant—is so-called because it is often a compound that can be used to support the induction of more than one AD model. CFA, for example, supports the induction not only of EAM, as in the previous paragraph, but several of the other AD to be discussed in this study. Adjuvants are formally defined as “nonspecific immune potentiators” and are thought to stimulate innate immune processes to produce the cytokines necessary to support AD. One result of the analysis that follows will be to question whether adjuvants are actually as “nonspecific” as they are thought to be in the context of AD.

Because several of the animal models of EAM (and the other AD to be discussed below) use well-defined, purified antigens and adjuvants, it is possible to examine which TLR are stimulated by each antigen or adjuvant in the production of the models (Table 1). Both LPS and coxsackievirus activate TLR4 and coxsackievirus also activates TLR2 [25]. Cardiac myosin and its streptococcal mimic M protein, in contrast, stimulate TLR2, TLR7 and TLR8 and no other TLR [27,29]. (For comparison, skeletal myosin activated no TLR [25].) Complete Freund’s adjuvant (CFA) activates TLRs 1, 2, 4, 9 and possibly 3 (though data are conflicting for TLR3) [30,31]. Thus, to fulfill the set of TLR that are activated in all three models of EAM (2, 4 and 7), and to mimic the TLR profile of human AM, both a source of TLR2 and 7 activator (cardiac myosin or streptococcal M protein) and a TLR4 activator (LPS or CFA) are necessary. Coxsackievirus activates mainly TLRs 4, 7 and 8 in human beings [22] while it [32] and CFA [27,29] also activate NOD2, which is upregulated in the human myocarditis and in the three animal models [33]. The common TLR4 and NOD2 activation by coxsackievirus confirms Fairweather, et al.’s [26] conjecture that the virus, which in most AD theories would be the “antigen”, plays the role of an adjuvant such as LPS or CFA in the induction of EAM. (This viral activation of TLR4, by the way, is not common: for comparison, rotaviruses activate TLRs 2, 3, 7 and 8 [34].).

Unlike autoimmune myocarditis, myasthenia gravis (MG) and its models require activation of TLR3 and TLR9, but not TLR2 (Table 1) [35,36] and no NOD or NLR involvement has been reported for either patients or in animal models (although this may be because no one has yet looked for it). The standard animal model of experimental autoimmune MG (EAMG) utilizes CFA with purified acetylcholine receptor (AChR), the CFA providing TLR4 and TLR9 activation [35,36] while the AChR provides TLR7 and TLR8 activation [37]. The activation by AChR of TLRs 7 and 8 is consistent with the observation that Epstein-Barr virus, which may be a trigger of MG and expresses an AChR antigenic mimic [38], also stimulates TLR7 and TLR8 activity. A combination of two “adjuvants”, poly-I:C, a TLR3 agonist, with LPS, a TLR4 agonist, can replace the CFA resulting in an animal model in which TLRs 3, 4, 7 and 8 are activated [35,36]. However, TLR9 activation is found in human MG, this latter animal model suggests that it is not absolutely necessary for induction of autoimmunity directed at AChR.

Multiple sclerosis (MS) has, once again, a different TLR activation profile than AM or MG. MS results from an autoimmune attack on central nervous system myelinated nerves, destroying their myelin sheaths. The TLR activation profile in MS consists of 2, 4, 9 along with NOD2 [39]. Animal models of MS fall under the general name of experimental allergic encephalomyelitis (EAE), which has a number of differences from the human disease but is characterized by similar TLR profiles (Table 1). EAE animal models generally use one of several myelin antigens such as myelin basic protein (MBP) or myelin oligodendrocyte glycoprotein (MOG) in combination with an appropriate adjuvant. MBP can be combined with CFA, in one formulation, which activates the same TLR as are observed in MS itself [39]. CFA, which activates TLRs 2, 4 and 9, may be replaced with cytosyl-*p*-guanosyl oligodeoxynucleotides (CpG-ODN), an adjuvant that activates TLR9 in combination with LPS, at TLR4 activator [40,41]. Alternatively, a species-appropriate MBP peptide can be used in combination with the peptidoglycan *N*-acetyl-muramyl-D-Ala-isoglutamine, also known as muramyl dipeptide (MDP), which is a TLR2, 4-NOD2 activator [42,43,44]. This combination appears to activate TLRs 2, 4 and 9 and NOD2 [45,46], again satisfying the general innate activation motif of MS. Alternatively, EAE may be produced by combining MOG, a TLR2 activator, with CFA (TLRs 2, 4 and 9 and NOD1 and 2) [47] or MOG with CpG-ODN, an adjuvant that stimulated TLR activity to give a TLR4-TLR9 combination [48]. Note that in none of these models does TLR7 or TLR8 play a role as they do in both AM and MG.

Finally, rheumatoid arthritis has, yet again, a TLR activation profile distinct from the previous three AD. RA itself is characterized by activation of TLRs 2, 4, 5 and 7 [49,50] and NOD2, but not NOD1 [51]. Several animal models of RA exist, each of which mimics the same TLR/NOD pattern of activation. One is inoculation of streptococcal cell wall antigens into the joints of rabbits, which results in TLR2, 4 and 5 activation as well as NOD2 activation; activation of NOD1 inhibits development of this disease model [50,52,53]. Another common RA model induces the disease in mice or rats using a combination of collagen II (a TLR5 activator [54]) in combination with complete Freund’s adjuvant, which supplies the TLR2 and TLR4 and NOD2 activation seen in the model [50,55,56]. Last, there is a collagen-II-lipopolysaccharide model that is able to induce RA symptoms with only TLR4 (LPS) and TLR5 (collagen) activation [54].

Four points emerge clearly from Table 1. First, each AD characterized here has a different TLR/NOD profile. TLR3 activation is necessary in MG but antagonizes AM and MS. TLR7 and 8 are activated in AM, RA and MG, but not MS, while TLR9 is activated in MS, but not in MG or RA or in AM where it antagonizes the disease. NOD2 is activated in AM. MS and RA, but not MG.

Second, no single TLR or NOD is activated in every AD. There does not, therefore, appear to be any particular pattern of innate activation that is necessary or sufficient to induce or maintain all AD pathogenesis.

Third, all AD and their animal models seem to require multiple TLR and/or NOD to be activated. This point raises an important question of whether this characteristic can be generalized to all AD and, if so, why AD should require activation of multiple TLR/NOD and whether these multiple TLR/NOD activations fall into clear sets or classes. This point will be taken up again below.

Fourth, every disease in Table 1 (and most of the animal models) involves activation of both cell-membrane associated TLR (2, 4 or 5) and cytosolic NOD along with activation of one or more endosomal TLR such as TLRs 3, 7, 8 and/or 9. This point is interesting since it suggests that both bacteria-focused innate stimulation (cell-membrane TLR and NOD) and virus-focused innate stimulation (endosomal TLR) are often present simultaneously to support these AD. It will be interesting to determine whether this pattern extends to other AD and if so, why.

Moreover, fifth, no experimental AD can be induced with a single, well-defined “antigen”. An “adjuvant”—or perhaps, in light of the previous points—“second agent” or “second antigen” is always necessary. More on this point below.

### 2.3. Limited Substitutability of One Adjuvant for Another in AD Induction Related to TLR/NOD Activation Profiles

Table 1 raises the issue of whether “adjuvants” in the induction of experimental AD are actually performing the function of “non-specific immune potentiators” as their name denotes. A noteworthy aspect of the need for different adjuvants such as CFA, LPS, poly I:C or CpG-ODN in the induction of EAM, EAMG and EAE is that adjuvants are not interchangeable within a given AD model and differ from one AD model to another. CFA seems to be one of the few adjuvants that can be used in multiple AD models, probably because of the wide range of TLR and NOD that its Mycobacterial components activate. However, even CFA is not a universal “adjuvant” for AD induction, and it can sometimes be replaced by one or more better-defined “adjuvants”. More specific adjuvants such as LPS, poly I:C and CpG-ODN are much more difficult to replace. The non-substitutability of one “adjuvant” for another must lead to questioning of whether these are truly “nonspecific immune potentiators” as their name implies—or whether they play more specific roles in AD pathogenesis.

To begin with, TLR and/or NOD activation appears to be a prerequisite to the induction of all animal models of AD. Replacing CFA with incomplete Freund’s adjuvant (IFA), consisting of the mineral oils without any bacterial component, results in failure to induce the AD in every animal model for which CFA is normally used. This is not just a failure of IFA. One group attempted to replace CFA in the induction of EAM with two adjuvants equally potent in terms of macrophage activation, but neither of these substitutes worked [57]. Both substitutes, Emulsigen and Montanide, are, like IFA, TLR-independent stimulators of Th2 immunity [58].

Similar negative results were found in EAMG [35] when CFA was substituted with alum (aluminum hydroxide), an adjuvant often used in human vaccines that activates monocyte activity independently of TLR or inflammasome signaling [59]. AChR plus TiterMax, another TLR- and NOD-independent immunopotentiator [60], also failed to induce EAMG [35]. Lacking appropriate TLR activation, the acetylcholine receptor antigen was unable to induce EAMG [35]. Thus, even though these adjuvants have demonstrated innate immune functions, they provided none of the TLR or NOD stimulation provided by CFA and were therefore unable to support EAMG induction.

Attempts to substitute adjuvants in EAE have generally met with similar failures. Despite the wide range of adjuvants used in creating models of MS, CFA (TLR2, 4, 9 and NOD1, 2) could not be substituted by poly I:C (a TLR3 agonist) or zymosan (a TLR2 agonist) in the MOG-CFA model of EAE [61]. Neither one fulfilled, in conjunction with MOG (at TLR2 activator) the TLR2, 4, 9 motif characteristic of EAE models. Similarly, it was not possible to induce EAE in the MBP-CpG-LPS model with either MBP-CpG or MBP-LPS [40,41]; both adjuvants (CpG plus LPS) were necessary to fulfill the TLR2, 4, 9 motif that characterizes the disease and is usually supplied by CFA (Table 1).

In sum, the role of “adjuvants” in experimental AD is not non-specific. On the contrary, each AD requires a specific set of TLR and NOD to be activated and only “adjuvants” that satisfy the TLR/NOD profile of that AD can support induction of the experimental disease. As I shall demonstrate below in Section 2.7, there are additional constraints on “adjuvant” structure and function that makes them much more specific to each AD then even their TLR/NOD profiles suggest.

### 2.4. Difficulties Explaining the Roles of Innate Immunity in Autoimmune Disease

The observation that TLR and NOD activation play essential roles in AD induction leads naturally to the question of how they do so and what the relative roles of “antigen” and “adjuvant” are in this activation. It would be beneficial here if we could turn to theories of AD pathogenesis to enlighten these issues, but unfortunately, only one existing theory of AD induction explicitly addresses the role of innate immunity during the initiation of AD and no theory explains how innate immunity helps maintain chronic AD pathogenesis [62]. Moreover, all major theories are incompatible in one-way or another with our current understanding of innate activation in AD. It is therefore worth a quick overview of these AD theories in order to identify the problems that need to be addressed in integrating the role of innate immunity into our understanding of AD pathogenesis. These AD theories are also a good introduction to the more comprehensive and integrated theory that I will introduce thereafter.

The major theories of AD induction at present are arguably the molecular mimicry theory, the anti-idiotype theory, the hidden antigen (or immunological privilege) theory, epitope drift and the bystander activation theory. In practice, these theories often overlap [62,63].

The molecular mimicry theory (MMT) (Figure 2) proposes that AD is initiated when the immune system produces an adaptive immune response to a microbial or autologous antigen that is cross-reactive with one or more host antigens [64]. For example, various viruses including hepatitis B virus [64], Epstein-Barr virus and Human Herpes Virus 6 [65] display significant similarities to myelin proteins and are epidemiologically associated with MS. The M protein of Group A Streptococci (GAS) mimics cardiac myosin [66], while coxsackie B viruses mimic cardiac actin [67] and both microbes are epidemiologically associated with onset of AM [68]. The major problem with MMT is that no one has managed to find a pathogen expressing a molecular mimic capable, by itself, of initiating any AD. Animal models (see Table 1 and discussion above) always require TLR/NOD activation using a so-called “adjuvant” for which the theory has no explanation. In fact, less than one percent of people infected with the putative infectious triggers of any given AD go on to develop that AD, which cannot be explained within the theory or even by known rates of genetic predisposition [62,63]. Thus, MMT must assume that some defect in the development of tolerance permits cross-reactivity to result in active autoimmunity in select individuals but does so without specifying any mechanism. This difficulty is compounded by the observation that almost every pathogen and commensal microbe expresses proteins that are very similar to human proteins [69,70] so that some explanation for why AD are not literally ubiquitous is needed.

The anti-idiotype theory (AIT) (Figure 3), in contrast to MMT, proposes that microbes induce AD by complementarity to the inducing antigens rather than by mimicry of them [71,72]. The microbe induces an adaptive immune response that in turn stimulates production of anti-idiotypic antibodies or T-cells. Since idiotypic antibodies or T cells mimic the cellular receptors used by the microbe to target particular host cells, their anti-idiotypes will mimic the microbe itself and therefore attack the cellular receptors utilized by the microbe to target its infection of a particular tissue or cell type. For example, anti-idiotypic autoantibodies to coxsackievirus that recognize the coxsackie and adenovirus receptor (CAR) in heart tissue are found in both AM and its coxsackievirus-induced animal model [73,74]. Like MMT, AIT has no explicit explanation for why infections with microbes that utilize human receptors is very common, but ADs are very rare. AIT, too, must posit (if only implicitly) that an error occurs in some people’s tolerance for their own antigens. Thus, while coxsackievirus and GAS infections are extremely common, AIT has no explanation for why AM is very rare. I return to this question below when considering the complementary antigen theory.

The hidden antigen theory (HAT) (Figure 4) of AD proposes that autoimmunity is initiated when the immune system comes into contact with autoantigens that are immunologically privileged [75,76]. This theory proposes that such hidden antigens are not “seen” by the developing immune system and therefore tolerance to them is not developed. If such antigens are exposed to the immune system by some type of damage, then an autoimmune response will ensue. Presumably (though not explicitly), the role of supposed infectious triggers of AD is to cause the damage to tissues harboring hidden antigens, thereby initiating the AD process. Various problems are not addressed by HAT, the most notable being that injury caused by mechanical damage such as surgery releases hidden antigens such as cardiac myosin and myelin-associated proteins. While antibodies against these proteins do appear after surgery, they do not cause tissue destruction or lead to AD [77,78,79,80]. Is there, then, some difference between injury to a tissue due to mechanical means and injury due to infection in terms of the resulting innate immune response? Additionally, while HAT applies well to multiple sclerosis antigens hidden within myelin and cardiac myosin in autoimmune myocarditis, it does not appear to be relevant to understanding autoimmunity targeted at acetylcholine in myasthenia gravis, insulin in type 1 diabetes, autoimmune thyroiditis or other AD that involve readily accessible extracellular autoantigens.

As noted above, the various AD theories are not necessarily mutually exclusive. For example, MMT and HAT can easily be synthesized by assuming that there is an extracellular autoantigen that mimics a pathogenic trigger of the AD and that the initial damage to the targeted cells then releases a hidden antigen that is more antigenic than the extracellular target. The immune system is now faced with two autoantigenic targets that are similar, but not identical, which is to say similar “epitopes”. If the hidden antigen is more antigenic than the extracellular one, then the immune response will be selected to favor targeting the hidden antigen in a process called “epitope drift” [81,82]. Figure 4 implicitly incorporates MMT and epitope drift into its depiction of HAT. Such epitope drift has been identified in many AD including, for example, AM in which the initial target of the autoimmune response appears to be extracellular matrix proteins such as laminin, collagen IV, and CAR mimic the hidden antigens that later drive ongoing disease, cardiac myosin and actin [83].

None of the theories just summarized requires the innate immune system to do anything more than antigen presentation to activate the adaptive immune system. None explains why so-called adjuvants are required to induce experimental models of AD in animals. None explains why different AD should be characterized by different sets of TLR/NOD activation (Table 1). None explains why—if these adjuvants are, as their definition maintains, nonspecific potentiators of the immune system—one adjuvant cannot be substituted for another in these models. Indeed, since the end effect of TLR and NOD activation is mediated either by the TRIF/TIRAP or the MyD88 pathways, why does it matter how which pathway is activated as long as it supports the proper Th1 or Th2 adaptive response?

There is only one AD theory that addresses any of these questions and it does so essentially by proposing that it does not matter how or by what the innate immune system is activated as long as it leads to a chronic inflammatory environment to support a robust adaptive response. The bystander activation theory (BAT) (Figure 5) explicitly invokes innate immunity as an essential, but non-specific element of AD induction. The bystander theory is based on the observation that activated T cells rapidly die after the triggering antigens are removed, whereas an unrelated infection can provide ongoing stimulation via cytokine production that maintains their viability [84,85,86]. Bystander activation is a process in which an infection or inflammatory process of some kind at or near a particular tissue results in significant upregulation of innate immune function attracting a localized adaptive immune response [84,85,86]. In this theory, initiation of an AD by the AD-triggering agent is supported by an unrelated bystander infection or immunological activator that creates local cellular damage thereby stimulating innate immunity to produce a chronic inflammatory environment. The concomitant over-activation of the adaptive immune response stemming from bystander and AD-triggering events breaks tolerance to host antigens.

BAT has been integrated explicitly with molecular mimicry as a means for explaining how tolerance to self-antigens is broken [87,88], but is equally amenable to explaining why some people develop anti-idiotypic adaptive immune responses (the bystander activation leads to over-production of the idiotype antibody leading to anti-idiotype production) or how hidden antigens become revealed to the immune system (the bystander infection causes local cellular breakdown, releasing the hidden antigens). For example, it has been found that adding CFA or LPS to idiotypic antibody inoculations induces animals to produce anti-idiotypic antibodies with far greater efficiency than if such adjuvants are omitted [89,90]. Presumably, the effect of these adjuvants is to act as bystander activators ramping up the production of cytokines to support the immunogenicity of the idiotypic antibodies.

BAT ignores several phenomena already described in this study such as the requirement for specific TLR and/or NOD activation. First, the theory does not explain how chronic inflammation is maintained to support chronic AD after the bystander infection is resolved. Second, if adjuvants are just nonspecific immune potentiators standing in for bystander activation by microbes, then why is it not possible to use truly nonspecific adjuvants such as incomplete Freund’s adjuvant (lacking mycobacteria), alum, Emulsigen or Montanide as activators of AD? What makes the active adjuvants associated with each AD non-replaceable by adjuvants that activate different TLR and NOD? Finally, BAT does not explain AD pathogenesis does not follow any localized infection. In practice BAT requires one of the other AD mechanisms to be at work simultaneously.

In sum, most current theories of AD induction do not explicitly have a role for innate immunity, and none addresses the need for specific TLR and NOD sets to be triggered to induce and maintain a chronic inflammatory state in experimental (and probably human) AD. All existing theories are predicated on the assumption that some error in immune tolerance must occur to permit AD to develop and that such errors are rare since the probable triggers of AD make use of mechanisms that should be very common.

### 2.5. Complementary Antigen Theory of Autoimmunity

A way out of this seeming impasse may exist by extending an integrative AD theory that I previously proposed. My complementary antigen theory (CAT) (Figure 6) starts with an assumption that runs contrary to all existing AD theories by asserting that loss of tolerance in autoimmunity is not a result of some “defect” in either the adaptive or innate immune systems, but rather a normal response to unusual circumstances. The unusual circumstances consist of encountering specific pairs of combined infections, in the case of AM, for example, a combined coxsackievirus–GAS infection. While both infections are common, the number of people encountering overlapping infections is extremely small and therefor the number of people at risk of AM, also very small. CAT is based on the assumption that the immune system evolved to handle one infection at a time and can be “confused” when presented with multiple concurrent infections.

A clarification is perhaps in order here concerning what is meant by “multiple concurrent infections” since it may be assumed that the microbiome represents an exemplary case of such concurrent infections. I use the term “infection” to mean something that activates the immune system to eliminate it. A normal microbiome does not satisfy that definition of infection. However, the immune system constantly samples the constituents of the microbiome, it very rarely produces an active response to it because it is able to distinguish between commensal microbes and pathogens at both the innate and adaptive levels (reviewed in [91]). Microbiome constituents have apparently been selected for their antigenic similarity to “self” antigens, including T cell receptors, so that microbiome constituents are treated as “self” by the immune system [69,70]. Because of this immune system mimicry, a healthy microbiome appears to act like a secondary immune system, controlling potential pathogens [91,92,93,94,95]. Thus, the microbiome does not qualify as something that can activate a CAT mechanism although elements of it may be targeted by AD if they mimic the host antigens attacked by the AD [69]. Not incidentally, the same factors explain why the host mimicry inherent in the microbiome does not stimulate AD through MMT or BAT.

CAT proposes that AD is induced when the immune system is stimulated by two complementary antigens simultaneously each of which induces an active immunological response and at least one of which mimics a host antigen [62,67,83,96,97,98,99,100]. The requirement for mimicry means that all evidence accumulated to support molecular mimicry in AD is also incorporated into CAT. Because the initiating antigens are complementary to each other, the resulting idiotypic antibody or T cell responses will be complementary to each other. Thus, although both immune responses are idiotypic, their relationship to each other will be equivalent to idiotype-anti-idiotype pairs. In this way, all the evidence supporting the anti-idiotype theory is also incorporated into CAT. CAT further proposes that at least one of the host antigens mimicked by one of the initiating antigens must be a cell-surface antigen. Thus, the autoimmune process can begin at the cell surface, impairing cellular integrity and releasing hidden antigens that can either be molecular mimics themselves or become major antigens due to epitope drift. This requirement incorporates all of the evidence that exists for the hidden antigen and epitope drift theories of AD. Finally, CAT incorporates evidence for bystander activation by explicitly including the necessity for at least two antigens (or microbes) for the induction of AD. However, CAT differs from the bystander theory in proposing that the two antigens or microbes must be related through molecular complementarity, whereas the bystander theory places no restrictions on the antigenic relationship between the AD trigger and the bystander infection. Thus, where the bystander theory puts no limitations on what causes the local inflammation supporting initiation of an AD, CAT predicts that only specific co-infections will support induction of any particular AD.

The result of the immune system having to process a concurrent pair of complementary antigens is to create a conundrum within the immune system that requires it to break self-tolerance or simply fail to respond to the antigens at all (Figure 6). Each of the complementary antigens induces a complementary antibody or T cell receptor (TCR). Because the antigens are complementary, the antibodies (or TCR) will also be complementary to each other. Moreover, each antibody (or TCR) will mimic one of the initiating antigens (Figure 6). Thus, the immune responses to complementary antigens requires the immune system to produce responses that mimic the causative agents thereby confusing self and non-self. Autoimmunity, from the CAT perspective, therefore, starts with a “civil war” within the immune system itself. If, in addition, at least one of the antigens mimics a host antigen, then the “civil war” within the immune system will extend through cross-reactivity with the host antigen(s) to host tissues.

Several consequences follow. One is that the civil war within the immune system will produce either circulating immune complexes (CIC) or perivascular cuffing (if TCR are involved). CIC, as noted above in Section 2.1, will initiate complement activation with concomitant destruction of associated tissue. CIC and tissue destruction will each drive further macrophage or dendritic cell activation (Figure 6). Even more important, since each antibody or TCR that is produced by the complementary antigens mimics one of these antigens, these antibodies and/or TCR will continue to activate the innate immune system to produce more cytokines, thereby chronically driving the chronic inflammatory environment through a positive feedback loop.

Thus, CAT uniquely provides not only a “natural” and perhaps even necessary means of breaking tolerance in the induction of AD, but also explains how the adaptive immune response continues to drive innate immune response following that induction by mimicking the initiating antigens. Moreover, this mechanism only works if there are pairs of complementary antigens, as evidenced above from studies of TLR/NOD activation profiles and studies of antigen-“adjuvant” binding complexes. Finally, CAT also explains why AD are very rare since combined infections with appropriate antigenic complementarity will be rare.

For example, one can calculate the approximate odds of acquiring autoimmune myocarditis/rheumatic heart disease (AM/RHD) from the possibility that it is caused by a combination of a group A streptococcal (GAS) infection complicated by a coxsackievirus (CX) [67,68,83]. There are about 10–15 million symptomatic enterovirus infections each year in the US of which coxsackieviruses constitute about half [101]. There are about 2.5 million cases per year of GAS pharyngitis [102] and an equivalent number of cases of GAS skin diseases such as scabies, impetigo, etc. [103]. With a US population of about 350 million, the probability of a CX infection is therefore about 6/350 or 1/60 and of a GAS infection, 4/350 or 1/90. If AM/RHD is due to a combined infection, the probability of contracting such a combination is therefore 1/4200. Fourteen percent of the population has genes associated with increased RHD risk [104] so that the actual probability of contracting AM/RHD is about one seventh of 1/5400, or 1/38,000. The actual incidence of RHD in the US is estimated to be between 1/25,000 and 1/100,000 [105,106]. Among populations in which GAS and/or CX infections are more prevalent (e.g., impoverished people), this risk would obviously increase proportionally.

In sum, CAT predicts an incidence of AM/RHD within current estimates. This CAT prediction can be compared with the rates estimated from the other AD theories. The bystander activation theory (BAT) runs into the problem that if many infections in addition to coxsackieviruses are permitted to be bystander activators for GAS (or there are many additional activators for CX other than GAS), then the rate of AM/RHD should be many times greater than just calculated for the GAS/CX combination. MMT and AIT also predict far higher incidences of AM/RHD than are found. Correcting CX incidence of 1/60 and GAS of 1/90 by one-seventh (the genetic risk population) yields between 1/420 and 1/630 AM/RHD cases per year in the US, a risk that is two orders of magnitude greater than reality. In fact, worldwide, only about 1/5000 people infected with GAS develop RHD [102,107] and only 1/500 who contract CX develop AM [108]. There is, in short, no obvious way to make the risk of AM/RHD coincide with theories of causation except to posit specific combinations of infections that activate specific sets of innate and adaptive immune responses.

The key point in the current context of the role of innate immunity in AD pathogenesis is that CAT uniquely explains how the innate immune system plays not only a necessary supporting role in AD, but also a very specific one in each particular AD. In fact, when one examines the specific sets or classes of TLR and NOD that are activated in each AD and experimental AD model, CAT leads to further insights into how a chronic inflammatory state is initiated and maintained to support AD pathogenesis. The key resides in recent research concerning synergistic interactions between the various TLR and NOD and reveals that not only is there complementarity driving autoimmune pathogenesis within the adaptive immune system, but also within the innate system.

### 2.6. Synergisms with the TLR and NOD Signaling Pathways

Synergism is a super-additive effect resulting from activation of a positive feedback loop between two elements of an interactive system. Many synergisms have been documented within the innate immune system. These synergisms include TLR2 with TLR3, TLR2 with TLR4, TLR2 with TLR6, TLR 3 with TLRs 7 and 8, TLR 4 with TLR7, and TLR 4 with TLR 8 and 9 [109,110,111,112,113,114,115,116,117,118]. For example, TLR2 and 4 synergize in rheumatoid arthritis-derived synoviocytes [119,120], rheumatoid arthritis [120], sarcoidosis [121], systemic lupus erythematosus, systemic sclerosis, Sjogren’s syndrome, psoriasis, multiple sclerosis and autoimmune diabetes [120]. In addition, TLR3, TLR4 and TLR9 agonists each synergize with NOD1 and NOD2 agonists in dendritic cells to induce gamma-IFN and IL-12 [122,123,124,125].

TLR antagonisms, in which the combined effect is significantly less than predicted from individual activation patterns, seem to be much rarer than synergisms: TLR2 agonists block subsequent activation of TLR3 and TLR4, presumably through a downregulation mechanism [126]; TLR7 and 8 agonists similarly antagonize TLR9 agonists [111]. These results suggest that timing of TLR activation may be important: costimulation with TLR2-3 and TLR2-4 is synergistic (see previous paragraph), yet TLR2 followed by TLR3 or TLR4 is antagonistic. The simplest way to explain such a time-dependent effect is to assume a feedback system between these TLR sets such that turning both on simultaneously is super-additive while turning one on by itself turns down the other member of the set.

The existence of both positive and negative feedback loops between TLR (and perhaps NOD) that is time-dependent has important implications for understanding how activation of innate immunity can modify AD susceptibility and progression.

### 2.7. Complementarity of Antigens in the Induction of AD

Now, if CAT is correct, two related types of complementarity should exist between the antigens inducing any particular AD. One type of complementarity is between the stereochemical structures of the antigens so that each antigen induces an immune response (T cell receptor or antibody) that is complementary and therefore anti-idiotypic, to the other. The other type of complementarity is functional such that these antigen pairs activate synergistic sets of TLR and NOD. There is evidence that at least some of the triggers of experimental AD share both properties.

Stereochemical complementarity of antigens has been documented in a number of AD by two methods. The first involves direct studies of antigens binding to each other using physicochemical methods. The second involves studies of antibodies against such antigens that recognize each other as idiotype-anti-idiotype pairs and therefore also bind to each other.

A number of physicochemical studies have demonstrated binding between AD-associated antigen or antigen-“adjuvant” pairs. For example, one EAE model for MS employs a combination of myelin basic protein (MBP) or its encephalitogenic peptides in combination with muramyl dipeptide (MDP) in Lewis rats or various strains of mice [127]. MDP has been demonstrated to bind directly to MBP at a site that corresponds to one of its major encephalitogenic regions [128,129,130]. Another EAE model (Table 1) uses a combination of myelin oligodendrocyte glycoprotein (MOG) in combination with the polynucleotide “adjuvant” CpG-ODN [48]. MOG has been demonstrated to bind directly to CpG-ODN and similar polynucleotides to form stable complexes [131].

Similarly, one-way to induce autoimmune myocarditis (AM) is to inoculate animals with the M protein of group A *Streptococci* (GAS) in combination with FCA (Table 1); however, the FCA can be replaced with lipopeptides that form stable complexes with the M protein and these complexes are more antigenic than is the M-protein-FCA combination [132,133]. Another way to induce EAM is to inoculate mice with coxsackievirus (CVB3) complexed with its host cellular receptor, the coxsackie and adenovirus receptor (CAR). Inoculating mice with CVB3 alone produces no EAM, nor does inoculating the mice with CAR alone, but their combination results in greatly enhanced acute myocarditis and frequent EAM [134]. An additional method for inducing EAM to inoculate Lewis rats with cardiac C protein in combination with antisera to the protein. Animals inoculated with this antigen-antibody combination developed chronic autoimmunity and dilated cardiomyopathy (DCM), whereas animals inoculated with either the C protein or the antisera did not. Notably, peptide fragments of cardiac C protein in complete Freund’s adjuvant did not develop autoimmunity or DCM [135] once again demonstrating that one “adjuvant” could not replace another.

Additional examples of enhanced immunogenicity and autoimmunity associated with complementary antigen complexes are reviewed in [136]. These include a diphtheria toxin-antitoxin (antibody) complex that induces an autoimmune polyneuritis similar to Guillain-Barré syndrome [137,138,139], the induction anti-insulin antibodies in type 1 diabetes by an insulin-glucagon complex [140] and improved antigen presentation to macrophages and dendritic cells by heat shock protein-viral antigen complexes [141].

The second method for demonstrating antigen complementarity involves their immunological properties. As predicted by CAT, complementary antigens should induce complementary T cell receptors and/or antibodies bearing idiotype-anti-idiotype relationships towards each other. Substantial evidence exists for such complementary immune responses. Combined coxsackievirus-group A streptococcal (GAS) infections often precede autoimmune myocarditis (AM) [141,142,143,144,145]. Antibodies against coxsackieviruses recognize several myocarditis autoantigens including cardiac actin and collagen IV and anti-actin antibodies recognize coxsackievirus antigens; antibodies against GAS recognize several other myocarditis autoantigens including cardiac myosin, the coxsackie and adenovirus receptor (CAR) and anti-myosin antibodies recognize GAS antigens [67,83]. Three pairs of these antigens are well-known molecular complements: actin and myosin bind to each other in the formation of muscle contractile fibers; laminin and collagen IV bind to each other to form the basement membrane of cardiomyocytes; and coxsackievirus is complementary to its receptor (CAR). Not surprisingly, then, GAS antibodies are anti-idiotypic to coxsackievirus antibodies [67,83] implying that the two microbes express dominant antigens that are antigenically complementary to each other.

Similar complementary antigenic relationships have been documented between antibodies against putative microbial triggers for several other autoimmune diseases as well. One is idiopathic thrombocytopenia purpura, in which GAS mimics von Willebrand factor (Factor IX), the main target of autoimmunity in the disease, while cytomegalovirus (CMV) mimics glycoprotein 1b (GP1b), another coagulation factor that binds to (and is therefore molecularly complementary to) von Willebrand factor during clotting. VWF and GP1b are also antigenically complementary as demonstrated by the fact that monoclonal antibodies against the two clotting factors bind to each other with high affinity. Moreover, since GAS mimics VWF and CMV mimics PG1b, it is not surprising that GAS antibodies bind with high affinity to CMV antibodies thereby demonstrating the antigenic complementarity of some of their dominant epitopes [107].

Not only do such antigenic complementarities exist among type 1 diabetes (T1D) antibodies [140], these antigenic complementarities have been extended to human T cell receptor sequences (TCR) that also display idiotype-anti-idiotype relationships to each other. In any given human T1D patient, there are TCR that bind to insulin and a distinct set of TCR that bind to the insulin receptor and/or glucagon, both of which are molecular complements of insulin [146,147]. These TCR recognize each other as idiotype-anti-idiotype pairs suggesting that T cells in T1D may literally aggregate through TCR-TCR binding to form perivascular cuffs. Such “cuff” formation has been demonstrated in vitro by complementary T cell subsets in EAE [90].

In sum, for a number of AD with well-defined animal models and/or molecular targets, including two AD featured here—AM and MS—evidence exists that initiating antigens exhibit complementarity towards each other by binding directly to each other and/or by inducing immune responses that have idiotype–anti-idiotype characteristics. Notably, antigen presenting cells are known to process such antigen complexes differently than are their individual antigenic components (reviewed in [136]). Antigenic complexes are more stable than their components to proteolytic and oxidative degradation and thus more likely to be perceived by the immune system as antigenic; they are digested into different fragments by the proteasome than are their individual components; and such complexes can be perceived by the immune system as distinctly novel antigens that differ significantly from their individual components [136]. Thus, stereochemical complementarity between the antigens triggering AD has very significant effects on the resulting adaptive immune response.

### 2.8. Complementary Antigen Theory Integrated with TLR/NOD Synergisms

Antigen complexation also has very significant effects on the innate immune response. CAT predicts that a further consequence of antigenic complementarity as expressed through idiotypic-anti-idiotypic adaptive immunity will be stimulation of complementary innate immune mechanisms. Complementary antigen pairs should drive synergistic TLR/NOD pairs creating the chronic inflammatory environment necessary to trigger and maintain AD. Incorporating the data concerning TLR and NOD activation for the various AD summarized in Table 1 into what is currently known about TLR/NOD synergisms (Figure 7) demonstrates that complementary antigens do, indeed, drive synergistic innate immune responses.

Begin with AM. Autoimmune myocarditis can be induced by coxsackievirus combined with cardiac myosin or, as appears to occur in many human patients, with GAS bacteria expressing the cardiac myosin mimic, M protein (see above). These agents are known to activate TLRs 2, 4, 7 and NOD2 (Table 1 and associated text), which in turn engage in three sets of synergisms, TLR2-TLR4, TLR4-TLR7 and TLR4-NOD2 (Figure 8). In fact, AM itself is characterized by activation of these sets of synergisms as are all other EAM models (Table 1).

A similar story characterizes MS and its EAE animal models except that a different set of innate synergisms are at work. To start with one well-characterized EAE model, myelin basic protein (MBP) or its encephalitogenic peptides can be combined with complete Freund’s adjuvant (CFA)or its active components such as muramyl dipeptide (MDP) or lipopolysaccharide (LPS) (Table 1). As noted in the previous section, MBP and MDP are antigenic complements as is another EAE inducer combination, MOG–CpG–ODN. Each of these sets of agents are known to activate TLR2, TLR4, TLR9 and NOD2, resulting in four pairs of synergistic interactions: TLR2–TLR4; TLR4–TLR9; TLR4–NOD2; and TLR9–NOD2 (Figure 9).

Similar analysis for MG and its animal models based on Table 1 yields TLR3-TLR7 and TLR4-TLR7 synergisms, while RA (and its animal models) yields synergisms between TLR2-TLR4, TLR4-TLR7 and TLR2-NOD2 (Table 2).

Other AD express additional patterns of TLR/NOD synergisms. For example, TLR2, TLR3, TLR4 and NOD1 are activated in in both human patients and in animal models of Sjogren’s syndrome and autoimmune disease against cells producing tears, saliva and mucus [148,149] TLR7 and TLR9 are also implicated in the animal model though less certainly in human patients. These TLR/NOD activations result in the following synergisms (Figure 7, Table 2): TLR2-TLR3, TLR2-TLR4 and possibly TLR4-TLR7, TLR4-TLR9 and NOD1-TLR9.

Yet another set of TLR synergisms characterizes anti-phospholipid syndrome (APS), an autoimmune attack on red blood cells that results in abnormal blood clotting. In APS, there is no known NOD activation, but TLR1, 2, 4 and 6 are activated [150,151] resulting in TLR2-TLR4 and TLR4-TLR6 synergies (Figure 7, Table 2).

Finally, the TLR/NOD activation pattern in systemic lupus erythematosus (SLE), an AD that targets diverse organs and tissues in the body mainly through anti-polynucleotide antibodies, differs once again from the previous diseases. In SLE, TLR2, TLR4, TLR7, TLR9 and NOD2 are activated [152,153,154,155], leading to synergisms between TLR2-TLR4, TLR2-NOD2, TLR4-TLR7, TLR4-TLR9 and TLR4-NOD2.

In short, each AD is characterized by activation of multiple innate synergies involving pairs of inducing agents that are characterized by their innate immune system complementarity. In other words, antigenic complementarity results in innate immune system synergy. Within both the CAT and the BAT models, these combinations act on each other like bystander activators, though which is the “bystander”, and which is the “antigen” is not obvious within the context of synergy. Moreover, in addition to each AD being distinguished by distinct sets of TLR and NOD activation sets (Table 1), each of these sets results in further activation of distinct sets of innate synergisms (Table 2). Hence, once again, as in the case of TLR/NOD activation profiles, it will be interesting to determine whether sets of AD synergisms fall into distinguishable classes.

In sum, CAT can explain the specificities of the antigen pairs for supporting a given AD while BAT cannot; and CAT reveals specific stereochemical relationship between AD-associated antigens that link their adaptive and innate responses at a functional level through complementarity.

### 2.9. Antigenic Complementarity Drives Ongoing Innate Activation Throughout the Course of AD

CAT resolves one final problem regarding AD pathogenesis that remains mysterious from the viewpoint of other AD theories and that is how the activation of innate immunity is maintained chronically. Once the initiating antigenic triggers of the AD are eliminated by the adaptive response and, in particular, once so-called “adjuvants” or “bystander activators” are no longer present, one would expect that the innate response would disappear. However, this is not the case; chronic inflammation driven by innate activation continues throughout the course of AD. According to CAT, there are two main drivers of this chronic innate activation: circulating immune complexes and the complementary idiotypic adaptive immune responses causing the AD.

If one refers to Figure 6, it is apparent that one of the unique aspects of CAT compared with other AD theories is the necessary production of CIC,(in the case of antibody-mediated AD or, in the case of T-cell mediated AD, perivascular cuffing (aggregates of T and/or B cells expressing complementary or idiotype-anti-idiotype receptors). Such immune complexes are known to drive innate immunity. This process is particularly well-characterized for SLE, in which it has been shown that antibody complexes present RNA and DNA fragments to TLR7 and TLR9, driving their ongoing activation [156,157,158]. In SLE and other AD characterized by the presence of rheumatoid factor (aggregates of antibodies caused by anti-Fc antibodies), such as RA, TLR2 and TLR4 may also activated [159,160]. CIC activate TLR3, TLR7 and TLR9 in glomerular AD [161], TLR3-TLR7 and TLR3-TLR9 being synergistic pairs (Figure 7 and Table 2). Moreover, CIC activate TLR2, TLR3 and TLR4 in scleroderma [162], again resulting in two synergistic pairings, TLR2-TLR4 and TLR3-TLR4, that can continue to drive chronic inflammation. It is likely that CIC, which occur very commonly in AD, drive similar ongoing innate activation in most AD. Since increases in CIC correlate with increases in complement activation [18,19], if CIC (and/or perivascular cuffs) localize at the target tissues (as would be expected if their components bind to their autoimmune targets), then active tissue destruction will follow (Figure 10).

An additional source of innate-activating antigen complexes is complexes formed with heat shock proteins (HSP). HSP are released as a result of ongoing cellular stress or destruction during the course of an AD and can drive ongoing innate activation [163]. As noted above, HSP bind viral and other protein antigens producing stable complexes that are highly antigenic [136]. These HSP-antigen complexes activate a variety of TLR including TLR2 and TLR4 [164,165,166], which are notable not only for being very commonly activated in many AD (Table 1), but also synergistic in their combined effects (Figure 7 and Table 2). Thus, cellular stress caused by AD contributes to the ongoing pathogenesis of the disease through a positive feedback loop.

From the point of view of CAT, however, the most important chronic driver of innate immunity during AD has not been recognized previously and this is the adaptive immune response itself [136]. A logical consequence of complementarity between antigens in the initiation of AD is the production of antibodies and/or TCR that mimic the initiating antigens (Figure 6); each antigen gives rise to an idiotype that mimics the complementary antigen. This aspect of CAT has been well-validated for several AD antigen pairs and for their antibodies and TCR [67,83,90,91,92,93,94,95]. What has not been validated is the consequence, which is that these antigen-mimicking antibodies and TCR should activate the same TLR/NOD as the original antigens, producing the same sets of TLR/NOD activation and synergisms as the original antigens. This adaptive stimulation of innate immunity would produce the ongoing chronic inflammation driving continued adaptive autoimmunity through an effectively endless positive feedback loop. Notably, TLR autoantibodies are known to exist in SLE, anti-phospholipid syndrome and other AD [150,162,167], though whether these mimic the original antigens that triggered the disease is not known.

### 2.10. Implications of Synergistic TLR Activation for Therapeutics

Because of the importance of TLR and NOD for the initiation and maintenance of AD, significant time and energy has been put into developing antagonists to these receptors as possible therapies [168,169,170,171,172]. These antagonists range from nanoparticles and small molecules to modified adjuvants and ant-TLR antibodies, and from approaches that directly bind to the receptors to those that interfere with downstream signaling. For example, psoriasis is an AD directed against desmosomes in the skin and characterized by excessive activation of endosomal toll-like receptors TLR7, TLR8 and TLR9 [173]. A range of psoriasis therapies have been approved or are in current clinical trials that target these TLR that range from monoclonal antibodies to synthetic drugs to natural compounds [173]. In general, however, these potential therapeutics have had good preclinical efficacy in well-defined animal models but have failed during clinical trials [174,175]. Surprisingly, the only FDA-approved TLR-related drug, Imiquimod is, a TLR7 agonist rather than antagonist, although its efficacy may be mediated by its antagonism of TLR9 (see Figure 7 and associated text).

CAT may help to explain the therapeutic failure of TLR antagonists. By incorporating innate activation by synergistic complementary antigens, CAT adds a new perspective on how effective antagonism of TLR/NOD activation must be carried out. Because of the multiple TLR/NOD synergies associated with each AD, it may be necessary in all cases to target therapeutics at one or more of the receptors activated in each synergistic pair involved in each disease. One TLR antagonist is unlikely to be sufficient to interfere sufficiently with innate activation caused by multiple synergistic pairs. Thus, to break the positive feedback cycle maintaining the AD will require addressing all of the synergistic pairs involved in that disease. Moreover, as Table 1 and Table 2 demonstrate, no single TLR antagonist will be useful for treating more than a handful of AD because each AD (or at least each class of AD) is characterized by a different set TLR/NOD synergies. Thus, effective treatment of AD through the innate immune system may require sets if synergistic antagonists to combat the synergistic agonists that drive the disease process.

### 2.11. Unresolved Problems

Two important questions remain unresolved by CAT. One concerns the possible pitfalls inherent in of utilizing animal models to inform our understanding of human AD pathogenesis; the second, what level of ongoing stimulation by adaptive immune factors is required to maintain ongoing AD.

The animal-model issue can be broken down further into two problems, one concerning whether the doses of materials utilized to induce experimental AD are consonant with concentrations of microbial antigens present during the induction of human AD, and the second concerning differences between the TLR/NOD systems of animals such as mice and those of human beings. None of these questions have easy answers.

Most experimental animal models of AD are induced using known concentrations of antigen-adjuvant combinations delivered by inoculation in a concentrated form whereas human AD are associated with infections that produce unknown concentrations of antigens distributed throughout a tissue, organ or the entire body. Whether this difference in antigen distribution matters to the innate and/or adaptive responses appears to be unknown and may only be resolved fully if and when infectious models of AD induction can be compared directly to models produced using inactivated antigens derived from those infectious agents. However, some comparisons are possible that suggest that the concentrations of antigens involved in human AD are comparable or even significantly greater than those used in the induction of experimental AD. For example, doses of 100–500 μg of inactivated *M. tuberculosis* are often delivered in Freund’s complete adjuvant (FCA) as part of experimental AD induction [26,33,34,37,38,39]. In comparison, a dose of the mycobacterial vaccine BCG (Bacillus Calmette-Guerin) is delivered as 50 μg of the semi-dry mass of live BCG bacilli, which will then replicate to many times this mass. That 50 μg corresponds to about 500,000 colony forming units (CFU) (https://biomedlublin.com/en/produkty/szczepionka-przeciwgruzlicza-bcg-10/). As a further comparison, a tuberculosis patient has about 100,000 CFU or about 10 μg, of *M. tuberculosis*/gram of lung tissue [176]. Since a normal human lung weights about 800 g [177], this converts to 8 mg of *M. tuberculosis* in a human patient, clearly orders of magnitude more antigen than is used in experimental AD induction and certainly on a weight-for-weight or concentration basis, comparable. Similarly, 1000 plaque-forming units (PFU) of inactivated coxsackievirus (with associated cardiac myosin) is sufficient to induce EAM [178] while a hepatitis B patient has that many copies of HBV (or more) per milliliter of blood [179]. In short, antigen concentrations appear to just as high or higher, in human beings as in experimental animals, though further research is needed to determine how representative these examples are of antigen concentrations in other AD-associated infections.

Differences in TLR and NLR function between animals and humans may also affect our understanding of AD induction in ways that could turn out to be significant. TLR4 function differs between mice and humans, for example, and mice do not have a functional TLR10, but do have additional TLR that human beings do not [7,8,9,10]. In addition, our knowledge of TLR functions, activation and synergies is ever-evolving. Thus, while Table 1 and Table 2 and Figure 7, Figure 8, Figure 9 and Figure 10 provide an internally consistent view of TLR-NOD interactions as they are known at present and the data for animal models and humans is generally consistent, it is not possible to rule out significant surprises in the future.

Finally, CAT proposes that CIC and complementary idiotypic adaptive immune responses continue to drive AD pathogenesis by activating TLR and NOD after the initial antigens have been eliminated, but this mechanism leaves more questions than answers. For example, it is not known what the thresholds are for triggering chronic innate activation through CIC or idiotypic immune responses. It is not known whether these thresholds change while a chronic innate activation is being maintained or whether they increase, decrease or vary during the course of an AD. Resolving this question may help to resolve the thorny issue of why some AD are transient, others progressive, others variable or cyclic, and still others exacerbated by additional infections.

## 3. Summary with Tests and Predictions that Differentiate CAT from other AD Theories

To summarize, it has been demonstrated that TLR and NOD play essential roles in supporting AD induction and ongoing pathogenesis. Surprisingly, activation of no single TLR or NOD is associated with every AD. Rather, each AD is characterized by chronic activation of multiple TLR and/or NOD. Moreover, the sets of TLR and/or NOD activated in any given AD may differ from any other, though given the small number of TLR and NOD available, it is likely that TLR/NOD activation sets form classes that may prove useful for diagnosing and treating AD or for unraveling common disease mechanisms. Surprisingly, the sets of activated TLR/NOD in each AD (Table 1) always seem to involve multiple TLR-TLR and/or TLR-NOD synergisms (Figure 7 and Table 2). These synergisms help to explain the occurrence of the chronic inflammation underlying every AD. In addition, the particular sets or classes of TLR and NOD that are activated may explain whether any particular AD is characterized by a Th1 (cell-mediated) or Th2 (antibody-mediated) pathogenesis: Recall from the Section 2.1 above that TLRs 3, 4 and 6 can activate the TRIF pathway resulting in interleukin production and a Th1 (or cellular) immune response while the other TLR result in a Th2 response. Some AD will be characterized by a combination of both types of responses.

The theory presented here alters the concept of “adjuvant” within the context of AD induction. Activation of multiple TLR and/or NOD requires multiple antigens and, indeed, animal models of AD invariably employ either very complex antigens with multiple epitopes (e.g., CFA or streptococcal cell walls) or, much more commonly, two or more well-defined antigens, one often labeled as the “antigen” and the other the “adjuvant” (Table 1). However, it is demonstrated that these “adjuvants” do not fit the classic definition of an adjuvant, which is nonspecific immune potentiators. Instead, “adjuvants” in AD animal models have very specific relationships with their antigens, in many cases being both stereochemically complementary to the antigen as well as complementary in their innate activation. This makes the “antigen” and “adjuvant” co-equal in terms of their roles in inducing AD. It follows that human AD, like their animal models, probably require two or more complementary antigenic triggers working in tandem.

Of the most commonly cited theories used to explain AD, only BAT and CAT explicitly incorporate the need for more than one antigen to trigger an AD. While BAT is compatible with the other AD theories such as molecular mimicry (MMT) and anti-idiotype (AIT), these theories do not have an explicit role for innate immunity in their explanations of AD pathogenesis. BAT and CAT differ significantly one the crucial point of whether bystander activation of innate immunity is non-specific (BAT) or requires a specific, complementary relationship between the antigens inducing disease (CAT). The latter appears to be more in line with the evidence summarized in the previous paragraph. Notably, CAT also requires that at least one of the antigens inducing the AD must molecularly mimic a host antigen, thereby incorporating all the data thus far accumulated in support of MMT; it incorporates all of the data supporting AIT as well, since it requires the complementary antigens to produce immune responses that have an idiotype-anti-idiotype relationship to each other; and it is compatible with data supporting the hidden antigen (HAT) and epitope drift (ED) theories of AD as well.

CAT makes testable predictions that differ from any other AD theory. One is that any component of the antigen set required to induce, and AD can only be replaced by one that has an equivalent TLR/NOD activation profile. This prediction more precisely states that one cannot, for example, replace the “adjuvant” in an animal model of an AD that stimulates TLR4 with one that stimulates TLR3 or TLR6, nor can one do away with the TLR/NOD activation by replacing it with a TLR- or NOD-independent adjuvant such as alum or incomplete Freund’s adjuvant. CAT also predicts that the antigens involved in AD induction must activate complementary sets of TLR/NOD that are synergistic in their effects.

CAT makes the further prediction that the triggering antigens in AD will induce complementary immune responses such that each immune response mimics the complementary antigen (Figure 6). Because each of the antigens stimulates specific, complementary TLR, and because the immune responses (antibodies or TCR) mimic these antigens, the immune responses will continue to activate the initial antigen-driven TLR/NOD set throughout the course of the AD. One simple test of this prediction would be to induce antibodies against coxsackievirus in one mouse and antibodies against cardiac myosin in another and combine these antibodies in an inoculation into a third mouse. CAT predicts that the original mice will not experience either autoimmune disease or chronic inflammation whereas the third mouse will develop experimental autoimmune myocarditis and develop chronic inflammation driven specifically by TLR2, 4 and 7 and NOD2 (Table 1). Similarly, one could inoculate one Lewis rat with myelin basic protein and another with CFA and combine their sera in an inoculation into a third rat. The first rats should not develop EAE nor develop chronic inflammation whereas the third rat should develop EAE driven specifically by activation of TLR2, 4 and 9 and NOD2 (Table 1). Note that in neither case is the third animal exposed to the original antigens so that any TLR or NOD activation must be caused by the antibodies with which it is inoculated.

CAT also makes the unique prediction that antibodies involved in the induction of AD will, because they are complementary to each other, form circulating immune complexes involving direct antibody-antibody binding and that these immune complexes will be able to activate the TLR/NOD sets associated with that AD.

Finally, CAT predicts that suppression of AD by means of TLR/NOD antagonists will only succeed if at least one of every synergistic pair involved in the complete set of TLRs and NODs that are activated in that particular AD are antagonized, since otherwise synergistic drivers of chronic inflammation will remain to continue driving the adaptive attack on the host.

## Figures and Tables

**Figure 1 ijms-21-04645-f001:**
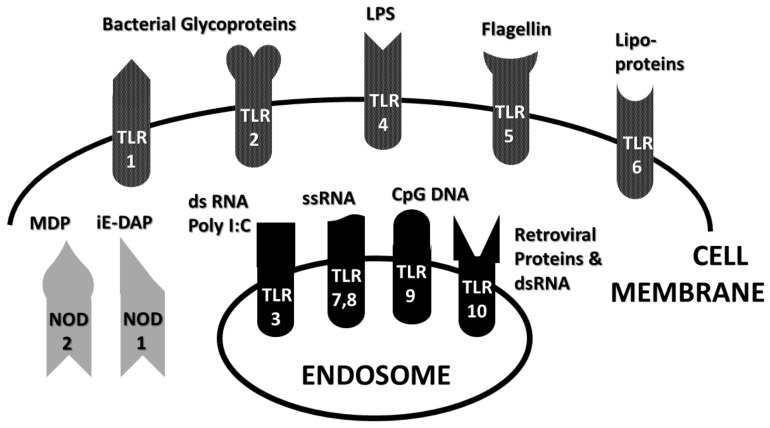
Schematic diagram summarizing the toll-like receptors (TLR) and nucleotide-binding oligomerization domain-containing proteins (NOD) locations (cell membrane, cytoplasm or endosome) on human innate immune system cells and their main activating ligands. MDP—*N*-acetyl-muramyl-D-alanyl-isoglutamine-containing peptides; iE-DAP—γ-D-glu-meso-diaminopimelic acid (iE-DAP) dipeptide-containing antigens; LPS—lipopolysaccharides; poly I:C—a polymer of inosine and cytosine that mimics double-stranded RNA (dsRNA); ssRNA—single-stranded polyribonucleic acids; CpG DNA—cytosyl-*p*-guanosyl oligodeoxynucleotide deoxyribonucleic acid; dsDNA—double stranded DNA.

**Figure 2 ijms-21-04645-f002:**
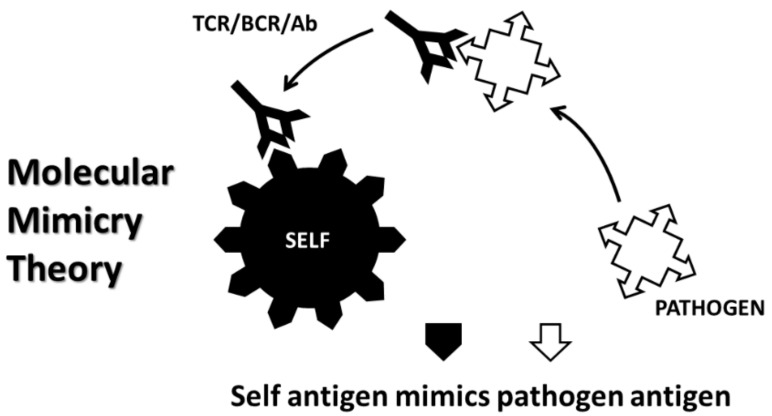
Schematic of the molecular mimicry theory (MMT). An pathogen or foreign antigen that mimics a host antigen initiates an immune response that results in a specific activation of a specific T cell receptor (TCR) an/or B cell receptor (BCR)—or B cell (antibody) response. Because the foreign antigen mimics the host antigen, the resulting immune response cross-reacts with the host antigen causing immune-mediated destruction of the target cell or tissue.

**Figure 3 ijms-21-04645-f003:**
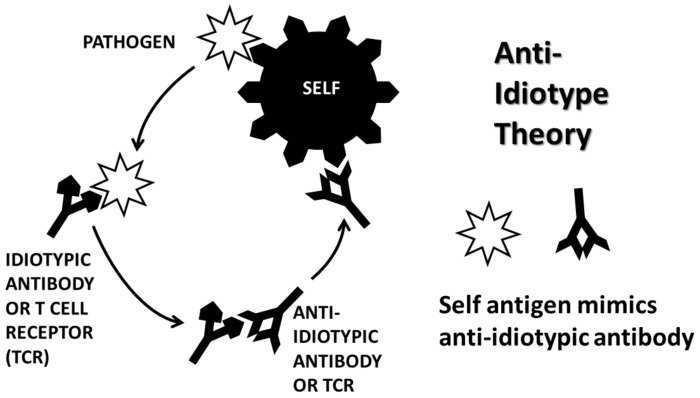
Schematic diagram of the anti-idiotype theory (AIT). AIT begins with the observation that many pathogens utilize cellular receptors or other extracellular proteins to direct their cellular tropism. If an immune response (antibody or T cell receptor-mediated) is directed at the receptor-binding antigen on the pathogen, then this idiotypic immune response will be complementary to the pathogen antigen and it will mimic the host cellular receptor. If the immune response to the pathogen is sufficiently robust, then enough idiotypic immune response (e.g., antibody) will be produced to activate an anti-idiotype. This anti-idiotype will be complementary to the idiotype and mimic the receptor-binding antigen of the pathogen. Because the anti-idiotype mimics the receptor-binding antigen, the anti-idiotype will recognize the same cellular receptor as the pathogen as being an immunological target, thereby initiating an autoimmune process.

**Figure 4 ijms-21-04645-f004:**
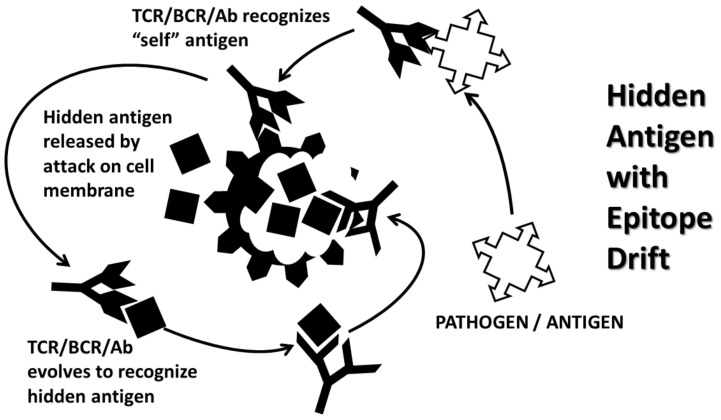
Schematic diagram of the hidden antigen theory (HAT) at work with epitope drift (ED). HAT can be initiated theoretically by any of three mechanisms. One would be mechanical or chemical harm to cells causing them to release hidden or “cryptic” antigens to which the immune system has not been tolerized during development. It may also be initiated by either MMT (Figure 2) or AIT (Figure 3) if cellular damage releases such hidden antigens. To illustrate how HAT may be integrated with other AD theories, it is illustrated here with MMT as the initiator followed by ED. The process begins with a pathogen or other antigen that mimics some extracellular host antigen. Antibody or T-cell mediated immunity cross-reacts with host cells bearing the antigenic mimic, releasing intracellular hidden antigens. These hidden antigens may be unrelated to the initiating pathogenic antigen and induce an entirely new immune response or the hidden antigens may also be mimics of either the pathogenic antigen or the original host mimic. In the latter case, the existing immune response may evolve to recognize more highly antigenic determinants (epitopes) on the hidden antigen, driving a more severe autoimmune response. The latter process is called epitope drift.

**Figure 5 ijms-21-04645-f005:**
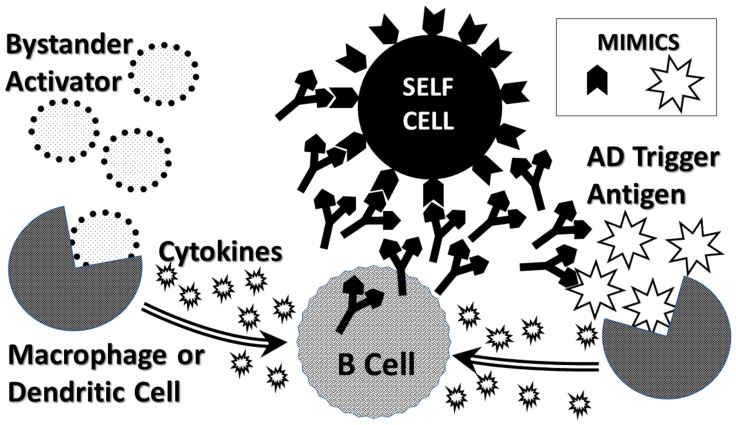
Schematic diagram of the bystander activation theory (BAT). BAT is the only current theory that explicitly integrates innate immunity into its explanation of pathogenesis. It assumes that there are two, independent sources of immunologic activation underlying initiation of any AD. One source of activation is a bystander infection or antigenic challenge that activates macrophages or dendritic cells to produce large quantities of cytokines. These cytokines draw additional monocytes to the area and stimulate any resulting T or B cell activity. Meanwhile, a second antigenic challenge, which will initiate the autoimmune disease (AD) occurs simultaneously. Once again, macrophages or dendritic cells process this AD trigger, initiating an appropriate T or B cell response. BAT is usually presented as an adjunct to MMT (Figure 2), but there is no reason that it cannot be compatible with any other AD theory (Figure 3 and Figure 4). The important aspect of BAT is that the simultaneous cytokine production by the AD triggering process along with the bystander stimulation results in hyper-activation of the B or T-cell mediated immune response to the AD trigger and, in consequence, overcomes tolerance to the host’s own antigen.

**Figure 6 ijms-21-04645-f006:**
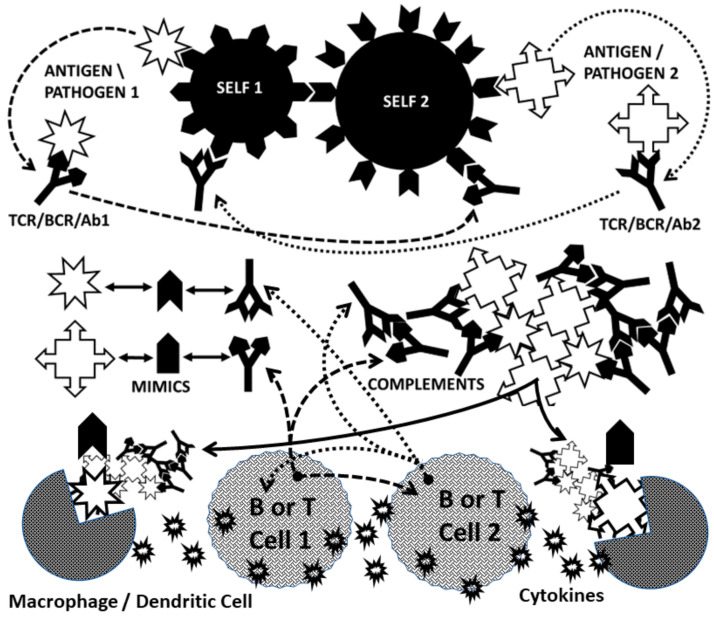
Schematic diagram of the complementary antigen theory (CAT). CAT integrates all of the AD theories described in Figure 2, Figure 3, Figure 4 and Figure 5 with important modifications. Instead of one antigenic trigger, CAT, like BAT, posits two. Unlike BAT, however, CAT requires that the triggering antigens be molecularly and antigenically complementary to each other. Each of these triggering antigens mimics a host (or self) antigen. These host antigens are themselves complementary to each other (though they may not directly interact as in this figure, which is drawn simply to illustrate their complementarity). As in MMT, Antigen 1 will elicit a complementary T or B cell response that will, in turn, recognize the host mimic of that antigen (labeled here as Self 2) as an autoimmune target. Concurrently, again as in MMT, Antigen 2 will elicit a complementary T or B cell response that recognizes the host mimic of that antigen (labeled here as Self 1) as an autoimmune target. It follows from the fact that the initiating antigens are complementary to each other, so their resulting immune responses are complementary to each other. In other words, although each immune response is idiotypic, they will have an idiotype-anti-idiotype relationship to each other, thus incorporating AIT. Finally, CAT integrates BAT in that the immune responses to both triggering antigens occur simultaneously so that each acts as a bystander activator for the other. Tolerance to host antigens in CAT is broken by the fact that immune system can no longer differentiate between triggering antigens, host antigens and its own idiotypic responses. The result is that the immune system not only attacks the triggering antigens and the host, but itself, producing circulating immune complexes (if antibody-mediated) and/or perivascular cuffs (if cell-mediated). This immunological “civil war” will continue to drive the AD long after the initiating antigens are gone because of the attacks of adaptive cells on each other (here illustrated by antibodies against complementary B cell receptors, but also possible between idiotypic and anti-idiotypic T cells) and stimulation of monocytes by immune complexes.

**Figure 7 ijms-21-04645-f007:**
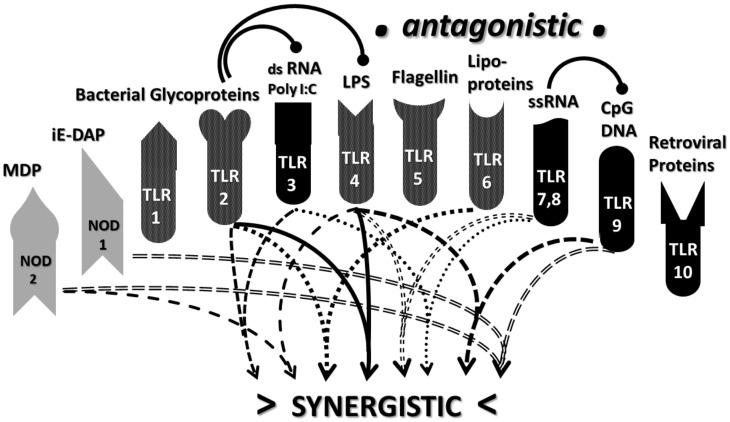
Schematic diagram summarizing toll-like receptor (TLR) and nucleotide-binding oligomerization domain-containing protein (NOD) synergistic and antagonistic interactions (see text for details).

**Figure 8 ijms-21-04645-f008:**
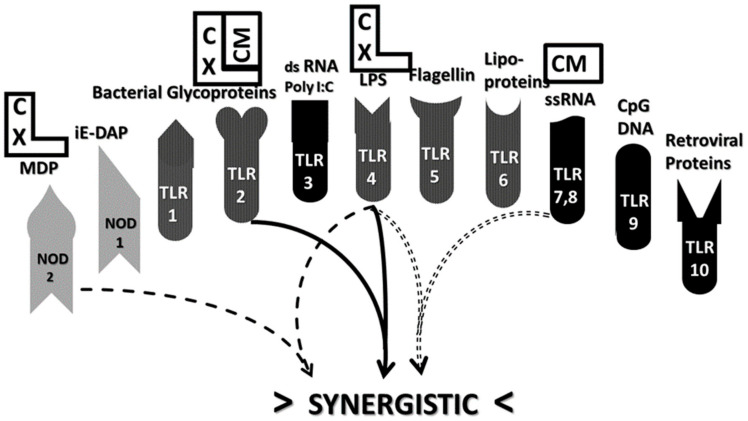
Schematic diagram showing the toll-like receptor (TLR) and nucleotide-binding oligomerization domain-containing protein (NOD) synergisms (Figure 7) that are expressed in autoimmune myocarditis (AM) and its animal model, experimental autoimmune myocarditis (EAM). These are illustrated here by the model using coxsackievirus (CX) and cardiac myosin (CM) as initiators, though the same sets of TLR/NOD activations occur in other EAM models and AM itself (Table 1 and Table 2). See Figure 1 for TLR and NOD-activator acronyms.

**Figure 9 ijms-21-04645-f009:**
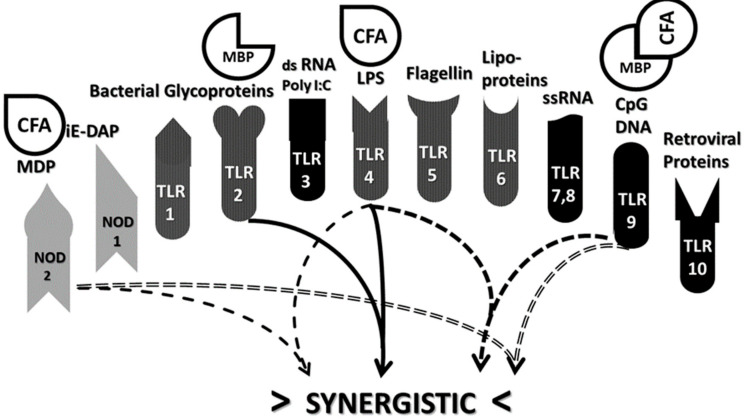
Schematic diagram showing the toll-like receptor (TLR) and nucleotide-binding oligomerization domain-containing protein (NOD) synergisms (Figure 7) that are expressed in multiple sclerosis (MS) and its animal model, experimental allergic encephalomyelitis (EAE). These are illustrated here by the model using complete Freund’s adjuvant (CFA) and myelin basic protein (MBP) as initiators, though the same sets of TLR/NOD activations occur in other EAE models and MS itself (Table 1 and Table 2). See Figure 1 for TLR and NOD activator acronyms.

**Figure 10 ijms-21-04645-f010:**
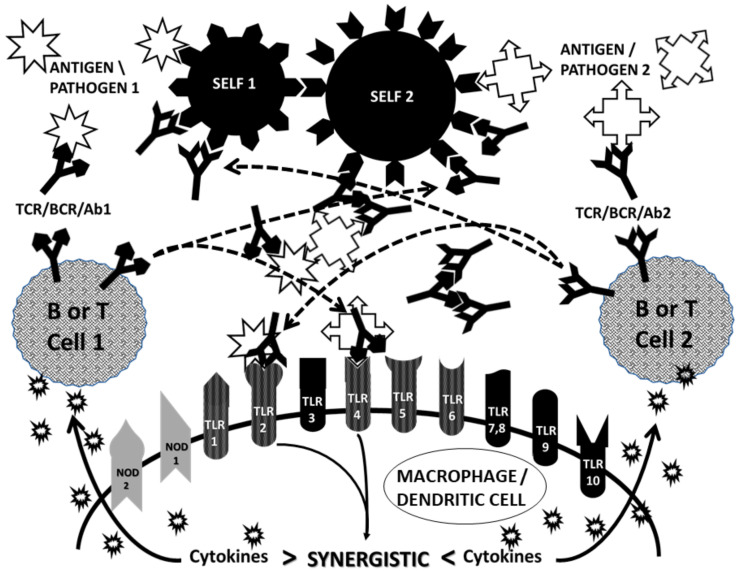
Schematic diagram of the complementary antigen theory (CAT) showing how it integrates (Figure 6 and Figure 7) synergistic TLR/NOD activation to create a chronic inflammatory environment supporting chronic autoimmune disease. Complementary antigens induce two sets of B or T cells to produce complementary immune responses (here illustrated as antibodies for simplicity) that mimic the antigenicity of the initiating antigens. The antigens and antibodies form circulating immune complexes that bind to target tissues mimicked by the antigens resulting in complement activation and cellular destruction. Since the antibodies mimic the antigens, they activate the same sets of TLR/NOD as did the initiating antigens (here illustrated by binding to TLR2 and TLR4), producing the same synergistic support of continued B (or T) cell activation on a chronic basis. In this way, complementary antigens break tolerance, induce chronic inflammation within the innate immune system by means of activating complementary, synergistic TLR/NOD and the adaptive immune system continues to drive its own upregulation through stimulation of innate immunity.

**Table 1 ijms-21-04645-t001:** Summary of toll-like receptor (TLR) and nucleotide-binding oligomerization domain-containing proteins (NOD) activation (X) in autoimmune myocarditis (AM), myasthenia gravis (MG), multiple sclerosis (MS) and rheumatoid arthritis (RA) and their most common animal models experimental AM (EAM), experimental autoimmune MG (EAMG), experimental allergic encephalomyelitis (EAE) for MS and experimental RA (see text for details and references). Most of the models are implemented in mice, but some are in rats or guinea pigs either instead or in addition. CX—coxsackievirus; CFA—complete Freund’s adjuvant; GAS M-prot—the M protein of group A *Streptococci*; AChR—acetylcholine receptor; I:C—poly I:C; LPS—lipopolysaccharide; MPB—myelin basic protein; MDP—muramyl dipeptide; MOG—myelin oligodendrocyte glycoprotein; CpG ODN—cytosyl-*p*-guanosyl oligodeoxynucleotides; Strep—*Streptococci*.

	TLR 1	TLR 2	TLR 3	TLR 4	TLR 5	TLR 6	TLR 7	TLR 8	TLR 9	TLR 10	NOD 1	NOD 2
**AUTOTIMMUNE** **MYOCARDITIS**		X		X			X	X				X
CX+myosin EAM		X		X			X	X				X
GAS M prot+CFA EAM		X		X			X	X				X
Cardiac myosin +CFA EAM		X		X			X	X				X
CX				X			X	X				X
Cardiac myosin		X						X				
GAS M protein		X						X				
CFA		X	?	X					X		X	X
**MYESTHENIA** **GRAVIS**			X	X			X	X	X			
EAMG (CFA+AChR)			X	X			X	X	X			
EAMG (I:C+LPS+AChR			X	X			X	X				
AChR							X	X				
Poly I:C			X									
LPS				X								
CFA		X	?	X					X		X	X
**MULTIPLE SCLEROSIS**		X		X					X			X
MBP-CFA EAE		X		X					X			X
MBP-CpG-LPS EAE		X		X					X			
MBP-MDP EAE				X					X			X
MOG-CFA EAE		X		X					X			
MOG-CpG EAE		X							X			
CpG ODN									X			
MDP		X		X								X
MBP		X							X			
MOG		X										
CFA		X	?	X					X		X	X
**RHEUMATOID ARTHRITIS**		X		X	X		X					X
Strep cell wall RA		X		X	X							X
Collagen II-CFA RA		X		X	X		X					X
Collagen II-LPS RA				X	X							
Collagen II					X							
LPS				X								
CFA		X	?	X					X		X	X

**Table 2 ijms-21-04645-t002:** Differing patterns of TLR and NOD synergisms associated with various autoimmune diseases based on Table 1 and Figure 7 as well as additional cases in the text. X indicates presence of that synergistic interaction. ? indicates that evidence for the synergism is questionable in light of current evidence (see text).

TLR & NOD SYNERGIES	TLR2-TLR3	TLR2-TLR4	TLR-TLR6	TLR3-TLR4	TLR3-TLR7, 8	TLR4-TLR7, 8	TLR4-TLR9	TLR3-NOD1, 2	TLR4-NOD1, 2	TLR9-NOD1, 2
Autoimmune myocarditis		X				X			X	
Myasthenia gravis				X	X	X				
Multiple sclerosis		X					X		X	X
Rheumatoid arthritis		X				X				
Sjogren’s syndrome	X	X				?	?			?
Anti-phospholipid syndrome		X	X							
Systemic lupus erythematosus		X				X	X		X	X

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
