# Peer review of "Synergistic Activation of Toll-Like and NOD Receptors by Complementary Antigens as Facilitators of Autoimmune Disease: Review, Model and Novel Predictions"

_ijms, 2020, doi:10.3390/ijms21134645_

Round 1

Reviewer 1 Report

This manuscript by Root-Bernstein reviews the role of the innate immune response in the initiation of autoimmune disease, with particular attention paid to how members of the TLR and NOD protein families are activated for signaling in different combinations.  The author reviews several theories of autoimmune disease induction before describing his own complementary antigen theory.  Overall, this was a persuasive review of the literature, and I only have a few minor points for consideration. 

  1. Line 36-38. Can this statement be matched to a statement of probability that one will suffer from an autoimmune disease in his or her lifetime?
  2. Is there a role for the microbiome in your CAT autoimmune model? If the immune system can only handle one infection at a time, how is it able to handle the microbiome-immune system interactions simultaneously? 
  3. Does it necessarily have to be a hyperinflammatory event that triggers AD? Is it hyperinflammation that causes all AD?  Is there a role for metaflammation (or other forms of chronic “low grade” inflammation)?  What are some of the metrics that are used to characterize the hyperinflammation that underlies every AD (paraphrased from line 827)? 
  4. There are some grammar issues throughout the manuscript that require some simple editing (lines 85-91, line 476, line 672, and elsewhere).
  5. Lines 113-115. Can the clinical and experimental studies be cited?
  6. In all of the animal models cited, there are ligands, antigens, and adjuvants that are being used to initiate the innate immune response that leads to AD. Are the dosages used in these models at concentrations that are physiologically relevant to humans?  Are they maintained at concentrations that are physiologically relevant to humans? 
  7. In Table 1, are all of the models described developed in mice? It would helpful to discern which model organisms are being used.  What are some drawbacks to using mouse models in developing this approach?  For example, there are differences in TLR4 function between mice and humans.  Another example is that TLR10 is not functional in mice, yet human TLR10 may heterodimerize with human TLRs, which may cause a more complicated picture.    
  8. The author describes two main drivers of chronic innate activation being CIC and complementary idiotypic adaptive immune responses. What are their thresholds for triggering chronic innate activation?  Do these thresholds change while a chronic innate activation is being maintained? 

Author Response

  1. Line 36-38. Can this statement be matched to a statement of probability that one will suffer from an autoimmune disease in his or her lifetime?
  1. Line 36-38. Can this statement be matched to a statement of probability that one will suffer from an autoimmune disease in his or her lifetime?

Yes! I have added the following sentence at line 38:  “The incidence of an AD should therefore be a function of the probability that a genetically susceptible individual will encounter such a synergistic set of antigenic stimuli. An adequate theory of AD should be able to calculate those odds.”

            I also return to this issue at LINE 363 and more completely at the paragraph added at LINE 583 where I provide an actual calculation based on combined GAS-coxsackievirus infection rates as a possible cause of AM/RHD.

  1. Is there a role for the microbiome in your CAT autoimmune model? If the immune system can only handle one infection at a time, how is it able to handle the microbiome-immune system interactions simultaneously? 

I have clarified my thinking on this point at a new paragraph starting at line 488 

  1. Does it necessarily have to be a hyperinflammatory event that triggers AD? Is it hyperinflammation that causes all AD?  Is there a role for metaflammation (or other forms of chronic “low grade” inflammation)?  What are some of the metrics that are used to characterize the hyperinflammation that underlies every AD (paraphrased from line 827)? 

Perhaps “chronic inflammation” would be a better description since the key point is not the degree of inflammation but the fact that stimulation of TLR/NOD does not cease when the initiating antigens disappear.  I have corrected my wording to reflect this conceptual change throughout the manuscript.

  1. here are some grammar issues throughout the manuscript that require some simple editing (lines 85-91, line 476, line 672, and elsewhere).

The manuscript has been re-read and these (and other) grammatical and spelling infelicities fixed.

  1. Lines 113-115. Can the clinical and experimental studies be cited?

They are, but further down in the text. I have clarified the line (“Clinical and experimental studies suggest the latter.”) by modifying it to read: “Clinical and experimental studies suggest  the latter as will be demonstrated in detail below and summarized in TABLE 1.”

  1. In all of the animal models cited, there are ligands, antigens, and adjuvants that are being used to initiate the innate immune response that leads to AD. Are the dosages used in these models at concentrations that are physiologically relevant to humans?  Are they maintained at concentrations that are physiologically relevant to humans? 

Excellent questions. I have added a discussion of these questions in a new Section 2.11 on outstanding problems.

  1. In Table 1, are all of the models described developed in mice? It would helpful to discern which model organisms are being used.  What are some drawbacks to using mouse models in developing this approach?  For example, there are differences in TLR4 function between mice and humans.  Another example is that TLR10 is not functional in mice, yet human TLR10 may heterodimerize with human TLRs, which may cause a more complicated picture.    

More excellent questions that I have added to my new Section 2.11 as well as in the caption to TABLE 1, where I indicate that most animal models are implemented in mice but some in rats or guinea pigs. 

  1. The author describes two main drivers of chronic innate activation being CIC and complementary idiotypic adaptive immune responses. What are their thresholds for triggering chronic innate activation?  Do these thresholds change while a chronic innate activation is being maintained? 

            Again, excellent questions that I have added to my new Section 2.11. For the previous questions.  While the previous questions have some data available that permits them to be addressed, I am not aware of any data pertinent to these questions, which makes them particularly interesting and important for future research.

Reviewer 2 Report

This is a comprehensive review on the role of TLR and NOD receptors in autoimmune diseases. The author is requested to take care of the following concerns.

  1. Throughout the manuscript, the author referred NOD-like receptor (NLR) as NOD. It is better to write NLR.
  2. Page 2 line 51-54: lymphokines are produced by T and B lymphocytes. Innate immune cells produce cytokines and chemokines. Please correct the relevant sentences.
  3. Page 2 line 61: PRRs recognize PAMPs and DAMPs. All PAMPs and DAMPs are not antigen. Please correct the sentence.
  4. Page 2 line 72: NOD like receptor activates different pathways including NF-kB and the inflammasomes. The sentence “ NOD activation leads to MYD88 activation and production of NF-kB is vague and incorrect.
  5. In pages 2 -3, it is mentioned that certain TLR induce Th1 response while other induce Th2 response. These descriptions are not very clear and no references were cited. TLR/TRIF/MyD88 pathways are mostly active in innate immune cells.  
  6. In figure 1, NOD1 and NOD2 are shown as membrane bound. However, NLRs are cytosolic proteins. Please confirm.
  7. Page 7 line 288: “ even those these adjuvants…..” please correct the sentence.
  8. Page 7 line291: “ the same story …….” not a meaningful sentence.
  9. Overall, the manuscript is too long. Better to shorten it if possible.

Author Response

This is a comprehensive review on the role of TLR and NOD receptors in autoimmune diseases. The author is requested to take care of the following concerns.

    Throughout the manuscript, the author referred NOD-like receptor (NLR) as NOD. It is better to write NLR.

                The Reviewer has evidently missed the sentence starting at line 88 about why I am using NOD rather than NLR :Because little work has been done on NLR more generally with regard to AD, I will focus mainly on the roles of NOD1 and NOD2 here [5].”  (See also line 64-65.)  I have attempted to emphasize this point by stating more clearly wherever possible that I am referring explicitly to NOD1 and NOD2, not to NLR in general.

 Page 2 line 51-54: lymphokines are produced by T and B lymphocytes. Innate immune cells produce cytokines and chemokines. Please correct the relevant sentences.

                FIXED

    Page 2 line 61: PRRs recognize PAMPs and DAMPs. All PAMPs and DAMPs are not antigen. Please correct the sentence.

                FIXED

    Page 2 line 72: NOD like receptor activates different pathways including NF-kB and the inflammasomes. The sentence “ NOD activation leads to MYD88 activation and production of NF-kB is vague and incorrect.

I believe the reviewer is correct in the general case of NLR;  but as indicated above, I am only discussing NOD1 and NOD2, which I have emphasized  by adding their numbers to the relevant sentence. With regard to those to specific NLR, I believe that the reference that I provided [4] and the following (which is now cited as well) both support my statement:        

Lilian O. Moreira and Dario S. Zamboni. NOD1 and NOD2 signaling in infection and inflammation. Front. Immunol., 08 November 2012 | https://doi.org/10.3389/fimmu.2012.00328

    In pages 2 -3, it is mentioned that certain TLR induce Th1 response while other induce Th2 response. These descriptions are not very clear and no references were cited. TLR/TRIF/MyD88 pathways are mostly active in innate immune cells. 

To begin with, I did cite a reference for this statement: reference [3], but I have also added the following reference: Dabbagh, K.; Lewis, D.B. Toll-like Receptors and T-helper-1/T-helper-2 Responses. Curr Opin Infect Dis. 2003 Jun;16(3):199-204. doi: 10.1097/00001432-200306000-00003.

    In figure 1, NOD1 and NOD2 are shown as membrane bound. However, NLRs are cytosolic proteins. Please confirm.

                Yes! Thank you catching this! FIXED

    Page 7 line 288: “ even those these adjuvants…..” please correct the sentence.

                FIXED

    Page 7 line291: “ the same story …….” not a meaningful sentence.

                FIXED

    Overall, the manuscript is too long. Better to shorten it if possible.

                Since the other reviewers did not indicate that length was a problem, I have left the length as it was.

Reviewer 3 Report

The review by Robert Root-Bernstein describes a literature review of Synergistic Activation of Toll-Like and NOD 2 Receptors by Complementary Antigens as Facilitators of Autoimmune Disease. The review is very rich and well written and structured with a lot of molecular information on the different mechanisms and TLRs and NOD pathways involved in four AD. This review is very structured with perfect English and figures and tables explaining each section
I give my favorable decision to be published in the current form

Author Response

I thank the Reviewer for their positive comments!